# The Camera System for the IceCube Upgrade

**Woosik Kang** [1,*] , **Jiwoong Lee** [1] , **Steven Rodan** [1] , **Carsten Rott** [1,2] and **Christoph Tönnis** [1] on behalf of the IceCube Collaboration

1   Department of Physics, Sungkyunkwan University, Suwon 16419, Republic of Korea; dlwldnddlw@g.skku.edu (J.L.); steven.rodan84@gmail.com (S.R.); rott@skku.edu (C.R.); ctoennis1988@gmail.com (C.T.)

2   Department of Physics and Astronomy, University of Utah, Salt Lake City, UT 84112, USA

\*   Correspondence: woosik.kang@skku.edu

**Abstract:** As part of a currently ongoing upgrade to the IceCube Neutrino Observatory, seven new strings will be deployed in the central region of the detector to enhance the capability to detect neutrinos in the GeV range. A main science objective of the IceCube Upgrade is to improve the calibration of the IceCube detector as a means of reducing systematic uncertainties related to the optical properties of the ice. A novel camera and illumination system, consisting of more than 1900 cameras, in 700 newly developed optical modules of the IceCube Upgrade, has been developed. A combination of transmission and reflection photographic measurements will be used to measure the optical properties of bulk ice between strings and refrozen ice in the drill hole, to determine module positions, and to survey the local ice environments surrounding the sensor module. In this contribution, we present the production, acceptance testing, and the plan for post-deployment calibration measurements with this camera system.

**Keywords:** IceCube Neutrino Observatory; detector calibration; camera system

## 1. Introduction

The IceCube Neutrino Observatory [1], located at the geographic South Pole, is the world's largest neutrino telescope. The observatory consists of a Cherenkov radiation detector of one cubic kilometre volume utilising the ultra-pure Antarctic ice [2] at depths between 1450 m and 2450 m. The in-ice detector is equipped with a grid of Digital Optical Modules (DOMs), each with a single downward-facing 10-inch photomultiplier tube (PMT). There is also a square kilometre air-shower detector on the surface of the ice [3] as well as a denser optical array in the centre of the active volume for detecting neutrinos with GeV energies [4]. The primary scientific objectives of the detector are measuring the high-energy astrophysical neutrino fluxes and determining the sources of these fluxes [5,6].

The deployment of seven densely instrumented strings into the central inner volume of the existing IceCube detector with newly developed optical sensors is currently being prepared as the IceCube Upgrade [7]. On each string, new optical sensors will be placed with a vertical separation of 3 m between depths of 2160 m and 2430 m below the ice surface. The new optical sensors are of three different types. The pDOM is designed based on the existing IceCube DOMs with improved electronics, the D-Egg [8] has two 8-inch PMTs, one facing upward and the other downward, and the mDOM [9] has 24 three-inch PMTs distributed to have nearly uniform directional sensitivity.

One of the goals of this upgrade is characterising the optical properties of the South Pole ice, the detector medium of IceCube, precisely and thereby reducing the uncertainties in directional and energy event reconstruction. To achieve the goal, novel calibration devices will be deployed as part of the IceCube Upgrade, including the IceCube Upgrade camera system, which will be a key component to the calibration campaigns. The camera systems [10,11] are designed to measure the optical properties of the ice in the vicinity of

each optical sensor and to obtain information of the position and orientation of the optical sensors in the ice.

## 2. Materials and Methods

To carry out the proposed surveys, the camera system utilises camera modules, which will capture the light signatures from illumination LED boards accompanying each camera module and pointing in the same direction. The system, integrated inside each optical sensor, can image the surroundings using reflected and transmitted light.

All three types of new optical sensors will be equipped with three pairs of camera-illumination modules. In the mDOM, two camera modules will be installed in the upper hemisphere pointing at 45° angle to the left and right of the vertical. The third camera will be located at the bottom of the mDOM facing downward. An additional illumination board will be placed at the top of mDOM to illuminate the refrozen hole ice. For the D-Egg, all three cameras are integrated on a ring-shaped structure in the lower half of the sensor, pointing straight out horizontally, spaced evenly with 120° between them. The integration method for the pDOM is currently being developed. The images of integrated camera systems in mDOM and D-Egg are shown in Figure 1.

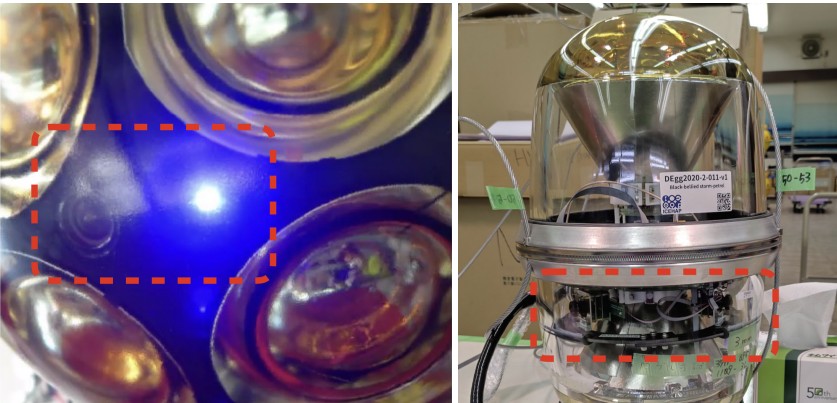

**Figure 1.** Images of the camera systems in the new optical sensors. Dashed red boxes indicate the cameras and LEDs. **Left**: A pair of cameras and LEDs for the mDOM that are integrated directly into the 3D-printed PMT holding structure. The cameras look through the glass of the pressure vessel via windows in the holding structure. **Right**: Cameras for the D-EGGs, which are attached to rings made from fibre-reinforced plastic (FR-4) using aluminium brackets. The rings are glued to the glass of the D-EGG pressure vessels using room-temperature vulcanizing silicone glue.

In Figure 2 on the left, the schematic illustrates surveying the refrozen hole ice with the downwards-facing camera in an mDOM. The camera captures direct and scattered light from an illumination module in the optical sensor below, and the optical properties of the refrozen ice will be determined based on the light distribution in the images. The other cameras on the mDOMs and D-EGGs measure the optical properties of the bulk ice between strings as shown in Figure 2 on the right. An LED on one of the optical sensors illuminates the surrounding bulk ice, and a camera in an optical sensor on a neighbouring string takes images of the light scattered through the ice. The optical properties of the ice can be inferred based on the distribution of incident light. As there are multiple cameras pointing in different directions, the bulk ice survey can be performed with directional dependence to measure the anisotropies in the optical properties of the Antarctic ice [12,13]. After the deployment, camera systems will only be operated during dedicated calibration runs. These runs will be kept short to minimize detector downtime, and multiple cameras and illumination LEDs will be operated simultaneously.

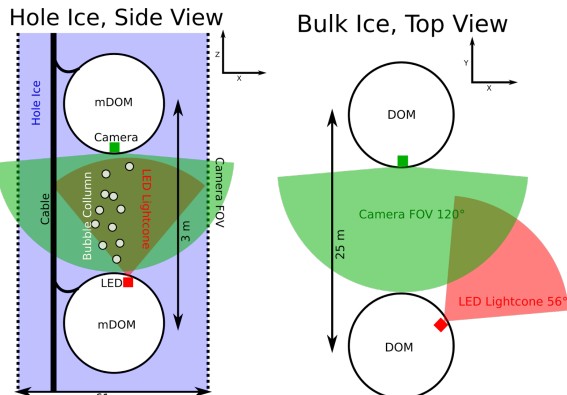

**Figure 2.** Schematic of planned measurements of the IceCube Upgrade camera system. **Left**: Refrozen hole ice measurement utilising two vertically separated optical sensors on the same string. A downward-facing camera observes light from an upward-pointing LED of the optical sensor below. **Right**: Bulk ice measurement utilising two optical modules on two different strings. A camera is observing scattered light from an LED on an adjacent string pointing at an angle of 60° to the camera. The schematics are not to scale.

To estimate the sensitivity and performance of the camera system in the Antarctic ice, dedicated simulation studies have been conducted based on a photon propagating Monte Carlo software [14], which was also used in previous camera studies [10,11]. The images in two different cases from the simulations are shown in Figure 3. The studies are not just to test the camera system in different measurement cases, but also to be extended to develop the image analysis techniques that will be used on the image data from the deployed system.

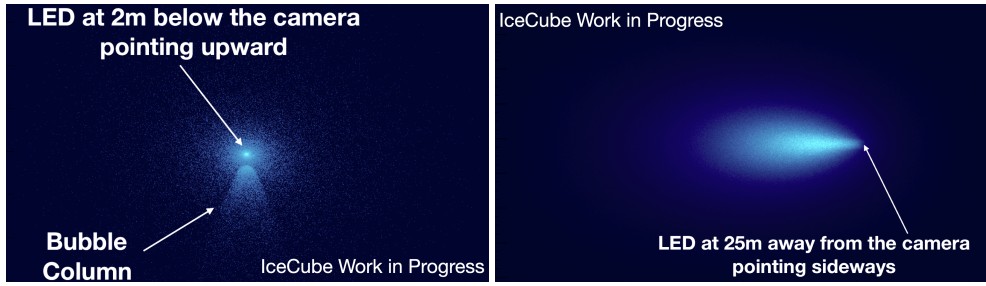

**Figure 3.** The expected camera images from the simulations. **Left**: the image represents visualising a column of ice with different optical properties than the surrounding ice, known as the 'Bubble column' in the refrozen hole ice. **Right**: the image shows the light cone from an optical sensor observed from another optical sensor on the adjacent string in the simulation.

## 3. Results

The production of the IceCube Upgrade camera system has been underway and is progressing steadily. For 278 D-Eggs, 1020 cameras were produced and have been integrated into D-Eggs. For 402 mDOMs, 1320 cameras are to be produced, and approximately 65% have been completed as of July 2022. For 14 pDOMs, 60 cameras will be produced. All numbers for cameras include the spares for tests and verification. The new optical sensors with the camera system are scheduled to be deployed at the South Pole by 2026.

The camera system has been tested at the South Pole as well as in a swimming pool. In the swimming pool test [10], the camera system demonstrated that it can image the object in the vicinity and resolve 10 cm separations at 25 m distance as shown in the top part of Figure 4. For the South Pole tests [15], one camera system was mounted inside the UV calibration logger [16] and deployed in the South Pole ice core hole during 2019/2020 Austral summer. After a successful deployment, a first analysis of the collected data has been conducted. Results are shown in the bottom part of Figure 4.

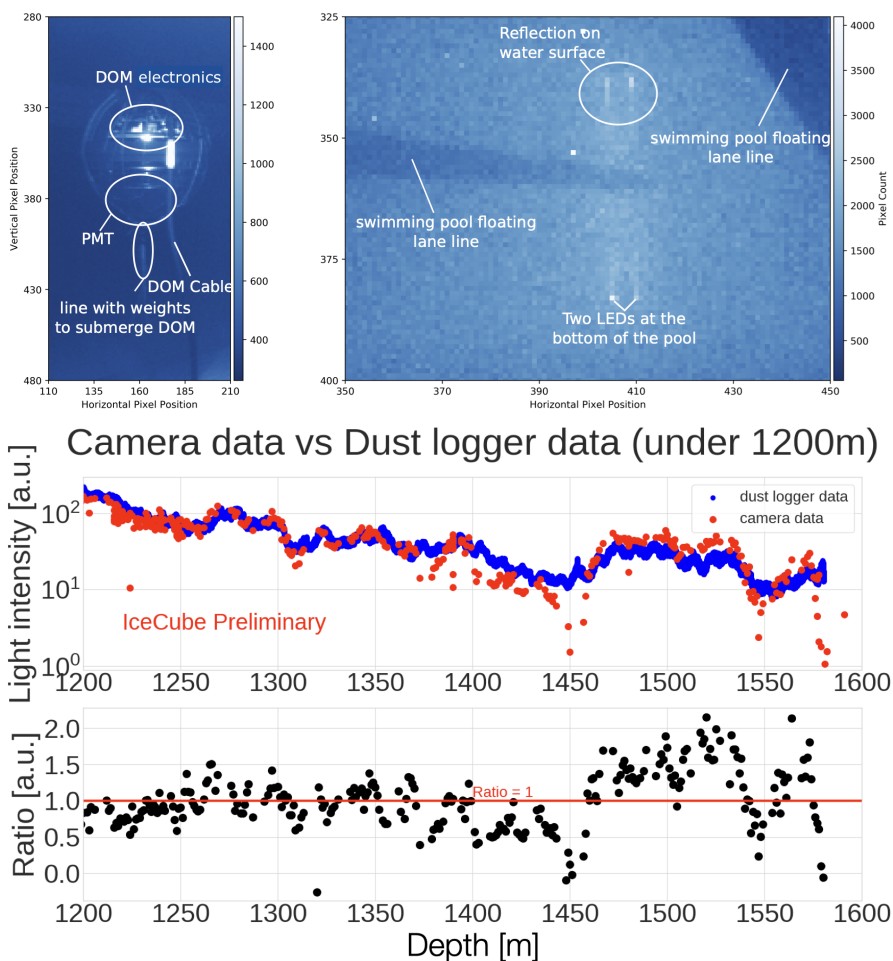

**Figure 4.** The results from camera tests at a swimming pool and at the South Pole. **Top**: the images taken in the swimming pool water. The left image shows an IceCube DOM at mid-depth at 2 m distance from the camera. The details of the DOM are clearly seen. The right picture represents two light beacons with 10 cm separation facing upward on the swimming pool floor at a distance of 25 m from the camera. The camera resolves the two beacons well. **Bottom**: the plots for comparison of the dust logger [17] and the camera brightness data as a function of depth between 1200 m and 1600 m. The upper panel gives the data points from each system and the lower panel shows the ratio of brightness between the camera data and the dust logger data. A detailed description is given in [15].

For the next-generation neutrino telescope at the South Pole, IceCube-Gen2 [18], a conceptual design of a camera system has been prepared. This camera system will be based on the experience from development and operation of the IceCube Upgrade camera system and would focus on the measurements on the refrozen ice in the drill hole. The images from back-scattered light will deliver information of the refrozen ice in the vicinity of each camera system. However, the inter-string measurements with this camera system will be limited because of the larger spacing between the Gen2 strings.

## 4. Conclusions

The cubic-kilometre photomultiplier array as realised in the IceCube Neutrino Observatory has been offering a unique insight into the properties of Antarctic ice. Since the construction of IceCube at the South Pole, intensive detector calibration campaigns have been carried out, which have delivered valuable information about the detector hardware as well as the detector medium, the Antarctic ice. The derived optical properties of the Antarctic ice are fundamental to the understanding of the detector and required for precise neutrino physics and neutrino astronomy measurements. In this aspect, the IceCube

Upgrade, which is now under construction, will provide an improved calibration of the detector. A novel camera system, which is going to be deployed as part of the programme, will be used to characterise the properties of the bulk ice and the refrozen ice in the drill hole with better precision using transmission and reflection images. Furthermore, for the next generation detector (IceCube-Gen2), a similar camera system is envisaged to be used for a comprehensive calibration of the detector medium, together with other calibration devices.

**Author Contributions:** W.K., C.R. and C.T. conceptualized the project. W.K. and C.T. developed the experimental test methodology. W.K., J.L. and C.T. developed the analysis software and the simulation framework. W.K and C.T. validated the experimental test methodology, the analysis software and the simulation framework. W.K. and J.L. performed the formal analysis and the simulation studies. W.K., J.L., S.R. and C.T. performed detailed investigations on the collected data and the simulation results. J.L. and S.R. managed the data curation. W.K., J.L. and S.R. visualized the results. W.K. wrote the original draft of this article. W.K., S.R., C.R. and C.T. contributed to the final version of the article by reviewing and editing. C.R. supervised and administrated the project and acquired the funding. All authors have read and agreed to the published version of the manuscript.

**Funding:** This research was funded by National Research Foundation of Korea grant number NRF2020R1A2C3008356.

**Institutional Review Board Statement:** Not applicable.

**Informed Consent Statement:** Not applicable.

**Data Availability Statement:** Not applicable.

**Conflicts of Interest:** The authors declare no conflicts of interest.

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
