# Peer review of "The Camera System for the IceCube Upgrade"

_psf, doi:10.3390/psf8010049_

Round 1

Reviewer 1 Report

The article reports on the development of an imaging system, comprising light sources and cameras, to be installed in the core of the IceCube neutrino observatory at the south Pole. Such system is meant be provide detailed characterization of the optical properties of the ice as well as a way to measure the positions and orientation of the optical modules. 

The article is carefully written and presents convincing results, from simulations and from tests in a pool and in situ, of the capabilities of the system.

Concerning language issues, I just noticed two cases of misspelling: "worlds" in place of "world's" at line 15 and "electronis" in the top-left image of figure 4.

Concerning the contents, I just noticed two small points:

* the paper contains little details about the technical implementation of the system (e.g., it is not even specified at which wavelength the LEDs emit) and of how the system was designed. It is understood that a detailed discussion of these topics is out of the scope of this article; however the authors might want to add some brief remarks or at least add references to previous studies at the end of the introduction; 

* using light sources, operation of the system seems incompatible with data taking of the apparatus. I think that a comment would be useful on how the system is expected to be used (with which frequency, and whether this is going to introduce a deadtime in data taking or not).

I have suggested consequently to do a minor revision of the text.

Author Response

Dear Reviewer1,

Thank you for your careful reviews and kind feedback. We have updated our manuscript with your comments, and below are our response to your points.

Point 1: Concerning language issues, I just noticed two cases of misspelling: "worlds" in place of "world's" at line 15 and "electronis" in the top-left image of figure 4.

Response 1: We checked the spelling and grammar of manuscripts again, corrected the misspelt words and improved the language properly. 

Point 2: the paper contains little details about the technical implementation of the system (e.g., it is not even specified at which wavelength the LEDs emit) and of how the system was designed. It is understood that a detailed discussion of these topics is out of the scope of this article; however the authors might want to add some brief remarks or at least add references to previous studies at the end of the introduction; 

Response 2: We added two references from previous studies in the Introduction section. The references contain detailed descriptions of hardware specifications, technical implementation, and mechanical integration of the system. You can find the references on line 36 of the revised version. 

Point 3: using light sources, operation of the system seems incompatible with data taking of the apparatus. I think that a comment would be useful on how the system is expected to be used (with which frequency, and whether this is going to introduce a deadtime in data taking or not).

Response 3: Explaining the operation plan of the system step-by-step is out of the scope of the manuscript. However, we have a concrete plan for system operation during and after the system deployment, and the outline of the system operation is now added in the updated manuscript on lines 64-66. The sentences suggest that the camera system will have dedicated calibration runs once a year after the deployment, and this will be kept short to minimize detector downtime.